# Integrated Spectroscopic Analysis of Wild Beers: Molecular Composition and Antioxidant Properties

**DOI:** 10.3390/ijms26146993

**Published:** 2025-07-21

**Authors:** Dessislava Gerginova, Plamena Staleva, Zhanina Petkova, Konstantina Priboyska, Plamen Chorbadzhiev, Ralitsa Chimshirova, Svetlana Simova

**Affiliations:** 1Institute of Organic Chemistry with Centre of Phytochemistry, Bulgarian Academy of Sciences, Acad. G. Bonchev str. Bl. 9, 1113 Sofia, Bulgaria; dessislava.gerginova@orgchm.bas.bg (D.G.); plamena.staleva@orgchm.bas.bg (P.S.); zhanina.petkova@orgchm.bas.bg (Z.P.); konstantina.priboyska@orgchm.bas.bg (K.P.); plamen.chorbadzhiev@orgchm.bas.bg (P.C.); ralitsa.chimshirova@orgchm.bas.bg (R.C.); 2Centre of Competence “Sustainable Utilization of Bio-Resources and Waste of Medicinal and Aromatic Plants for Innovative Bioactive Products” (BIORESOURCES BG), 1000 Sofia, Bulgaria; 3Laboratory for Extraction of Natural Products and Synthesis of Bioactive Compounds, Research and Development and Innovation Consortium, Sofia Tech Park JSC, 111 Tsarigradsko Shose Blvd., 1784 Sofia, Bulgaria; 4Faculty of Chemical and Systems Engineering, University of Chemical Technology and Metallurgy, 8 St Kliment Okhridski blvd, 1756 Sofia, Bulgaria

**Keywords:** metabolomics, beer, wild ales, NMR, LC-Orbitrap-MS, antioxidant activity, chemometrics

## Abstract

Wild ales represent a diverse category of spontaneously fermented beers, influenced by complex microbial populations and variable ingredients. This study employed an integrated metabolomic profiling approach combining proton nuclear magnetic resonance (^1^H NMR) spectroscopy, liquid chromatography–mass spectrometry (LC-MS), and spectrophotometric assays (DPPH and FRAP) to characterize the molecular composition and antioxidant potential of 22 wild ales from six countries. A total of 53 compounds were identified and quantified using NMR, while 62 compounds were identified by using LC-MS. The compounds in question included organic acids, amino acids, sugars, alcohols, bitter acids, phenolic compounds, and others. Ingredient-based clustering revealed that the addition of dark fruits resulted in a significant increase in the polyphenolic content and antioxidant activity. Concurrently, herb-infused and light-fruit beers exhibited divergent phytochemical profiles. Prolonged aging (>18 months) has been demonstrated to be associated with increased levels of certain amino acids, fermentation-derived aldehydes, and phenolic degradation products. However, the influence of maturation duration on the antioxidant capacity was found to be less significant than that of the type of fruit. Country-specific metabolite trends were revealed, indicating the influence of regional brewing practices on beer composition. Correlation analysis was employed to identify the major contributors to antioxidant activity, with salicylic, dihydroxybenzoic, and 4-hydroxybenzoic acids being identified as the most significant. These findings underscore the biochemical intricacy of wild ales and exemplify metabolomics’ capacity to correlate compositional variation with functionality and authenticity in spontaneously fermented beverages.

## 1. Introduction

Wild ales represent a distinctive and historical category of beer. The fermentation process utilized by these brewers is characterized by its reliance on a mixed-culture fermentation technique, which involves the simultaneous presence of wild yeasts and bacteria. This approach stands in contrast to the more commonly employed practice of controlled inoculations of *Saccharomyces cerevisiae* (the yeast commonly used in ales) or *Saccharomyces pastorianus* (the yeast typically used in lagers), which involve the deliberate introduction and subsequent cultivation of a specific strain of yeast [1]. The correlation between these beers and their environment is significant. It has been established that ambient microorganisms, which are frequently derived from the brewery’s microflora or the surrounding atmosphere, initiate spontaneous fermentation processes. This, in turn, contributes to the development of complex and diverse flavor profiles [2]. Fermentation in wild ales is characterized by protracted periods of fermentation, ranging from several months to years, and involves a succession of microbial populations, including *Brettanomyces* species and lactic acid bacteria (LAB), such as *Lactobacillus* and *Pediococcus*. These organisms thrive in low-pH, hop-rich environments and impart the characteristic sour, funky, and earthy flavors to wild ales [3]. Fermentation is conventionally undertaken in wooden barrels, where maturation serves to enhance the chemical and sensory intricacy of the beer. Another hallmark of wild ales is the use of non-conventional adjuncts, such as fruits, herbs, spices, and botanicals, which are added during fermentation or aging, thereby further diversifying their metabolite composition. Historically, wild fermentation was the standard before the isolation of pure yeast cultures in the 19th century [4]. Whilst the advent of modern brewing techniques has undoubtedly facilitated reproducibility and control, a significant number of craft brewers have now chosen to deliberately adopt wild fermentation, recognizing its capacity to yield flavor profiles that are both terroir-driven and unpredictable. The resurgence of wild ales can be seen as indicative of a broader trend in craft brewing that values authenticity, complexity, and traditional production techniques. This category encompasses a variety of distinct substyles distinguished by spontaneous or mixed fermentation, including lambic, coolship ales, gueuze, American wild ale, Flanders red ale, and oud bruin [2].

Notwithstanding the increasing production and popularity of wild ales, the correlation between microbial activity and chemical composition remains to be extensively investigated [5]. Recent research has concentrated on the microbial ecology of spontaneous fermentations, with the aim to identify how variables such as pH, temperature, and barrel aging shape microbial succession [6,7]. Furthermore, the evolution of key acids, ethanol, and residual sugars during fermentation with different yeast strains has been the subject of study, with the results highlighting the strain-specific fermentation kinetics and their influence on metabolite accumulation [8,9]. It is evident that characteristic compounds have been associated with specific microorganisms. For instance, 4-ethylguaiacol and 4-vinylguaiacol have been identified in *Brettanomyces*, lactic acid has been detected in LAB, 1,3-propanediol has been identified in *Clostridium* or *Bacillus* spp., and acetic acid has been found in Acetic Acid Bacteria (AAB) [10,11,12]. The compound, diacetyl, has been associated with the genus *Pediococcus*, while ethyl acetate, a common ester produced by *Saccharomyces* and *Brettanomyces*, has been shown to be influenced by Acetic Acid Bacteria [13]. While these compounds offer insight into microbial presence and activity, comprehensive comparative chemical analyses across wild ale substyles, ingredient types, and geographic origins are still rare. For instance, the understanding of the nuanced metabolomic differences between traditional lambics and American coolship ales has only recently begun to be developed [14]. In addition, while metabolomics has furnished substantial insight into the chemical evolution of wine during maturation, wild ales have not been the subject of a comparable, exhaustive investigation utilizing a chemometric framework [15]. A significant knowledge gap exists concerning the identification of the robust biomarkers of prolonged maturation, including oxidized bitter acids, advanced Maillard products, and evolved phenolic profiles. This is particularly true when accounting for confounding factors, such as fruit additions or the base beer style.

Most analytical studies conducted to date have concentrated on specific compound classes using gas chromatography–mass spectrometry (GC-MS), a technique that is particularly well suited for the analysis of volatile organic compounds (e.g., esters, higher alcohols, and volatile phenols). Whilst GC-MS has been demonstrated to elucidate the key aroma-active compounds associated with wild yeast metabolism, its scope is limited to volatile constituents [16]. In contrast, liquid chromatography–mass spectrometry (LC-MS) offers high sensitivity and extensive coverage of non-volatile and semi-volatile metabolites, encompassing organic acids, amino acids, and polyphenols derived from malt, hops, and fruit or herbal additions. Furthermore, it is frequently employed to identify hop-derived bitter acids (e.g., humulones and isohumulones) and their derivatives [17,18]. However, it should be noted that LC-MS methods frequently necessitate meticulous sample preparation and chromatographic separation. Moreover, these methods may be susceptible to matrix effects. In contrast, nuclear magnetic resonance (NMR) spectroscopy has been shown to provide rapid, non-destructive, and quantitative metabolite profiling with minimal sample preparation and strong reproducibility. Despite its reduced sensitivity in comparison to LC-MS when it comes to low-abundance compounds, NMR facilitates absolute quantification and the concurrent detection of multiple compound classes. These include alcohols, sugars, nucleosides, organic acids, aldehydes, and amino acids, and this is possible in complex matrices such as beer [19]. The combination of these two techniques has been shown to enhance the strengths of each individual technique, thereby providing a more comprehensive and detailed chemical profile than that which would be obtained by utilizing either technique in isolation [20].

Despite their chemical complexity, wild ales have not been thoroughly evaluated for their functional properties, such as antioxidant potential, in relation to their metabolomic profiles. The antioxidant activity of conventional beers has been extensively documented, and this activity is primarily attributed to polyphenols that are derived from malt and hops. Emerging evidence suggests that fruit beers and wild ales may exhibit enhanced antioxidant activity due to fruit-specific phenolics, and novel compounds formed during mixed fermentation [21]. However, systematic correlations between specific metabolite classes and antioxidant assays are lacking. A plethora of spectrophotometric assays, including DPPH (2,2-diphenyl-1-picrylhydrazyl) and FRAP (ferric reducing antioxidant power), have been employed to evaluate the total antioxidant capacity through discrete mechanisms [22,23]. The integration of these assays with metabolomics data facilitates the functional interpretation of chemical profiles. Nevertheless, their application in wild ale research remains underdeveloped. The considerable heterogeneity in wild ale production, stemming from differences in microbial consortia, ingredient use (especially fruits or herbs), and aging regimens, poses challenges to reproducibility and quality assessment. Metabolomics, in conjunction with chemometrics, provides a robust array of analytical tools with which to navigate this complexity. Supervised techniques, such as orthogonal partial least squares discriminant analysis (OPLS-DA), have been shown to be effective at identifying discriminant metabolites that explain group separations while controlling noise [24].

Metabolomics has been successfully applied to differentiate the geographic origin and aging profiles of other fermented products [25]. Nevertheless, its utilization in the domain of wild ale research remains constrained. There is an urgent need for systematic studies to be conducted exploring how fruit and herb additions, fermentation duration, and geographic origin influence wild ale metabolomes, and which biomarkers most strongly predict antioxidant activity.

The present study addresses these gaps by integrating NMR- and LC-MS-based metabolomic profiling with spectrophotometric antioxidant assays (DPPH and FRAP), supported by robust chemometric analysis. The objective of this study is to ascertain the influence of fruit and herb additions, aging period, and geographical origin on the metabolomic profile and antioxidant potential of wild ales. It is asserted that this will contribute to a more profound comprehension of their quality, functionality, and authenticity.

## 2. Results and Discussion

### 2.1. Molecular Composition of Wild Ales via ^1^H NMR Spectroscopy

The chemical profiles of 22 wild ales, all of which had been barrel-aged and produced in six different countries, were analyzed using quantitative ^1^H NMR spectroscopy with water suppression. The typical spectrum is presented in Figure 1.

A total of 53 metabolites were identified and quantified across all the samples (Table 1), encompassing alcohols, organic acids, amino acids, saccharides, and nucleosides.

As anticipated, ethanol was the predominant compound (average: 41,680 mg/L), followed by lactic acid (5915 mg/L), acetic acid (1749 mg/L), glycerol (1651 mg/L), and maltodextrin (1598 mg/L). The elevated concentration of lactic acid observed in this study aligns with previous findings for lambic and gueuze—substyles of wild ales—where the values range between 3670 and 17,470 milligrams per liter [26]. This finding provides further evidence to support the hypothesis that lactic acid bacteria (LAB) and Brettanomyces spp. play a significant role in the fermentation process [14]. The substantial levels of acetic acid detected suggest that secondary fermentation has occurred, driven by acetic acid bacteria, which are frequently active during the barrel-aging process. An analysis of the glycerol levels in wild ales and those in herb-infused styles, such as blanche beers, revealed that they were comparable [27]. However, methanol (60 mg/L) and 2,3-butanediol (223 mg/L) were significantly elevated. Notably, the methanol content in certain samples was found to exceed the levels typically observed in red wines, while the butanediol concentrations were comparable to those reported in grape ales (~206 mg/L) [19]. It can be posited that these trends may be indicative of the microbial diversity in spontaneous fermentation, a high amount of pectin and its demethylation or of specific metabolic pathways that are favored during extended aging [28]. It is noteworthy that 1,3-propanediol, a compound that is otherwise uncommon in beer, was detected in a number of samples. The presence of this substance has been linked to the activity of glycerol-fermenting bacteria, such as *Clostridium* or *Bacillus* spp., which are known to convert glycerol to 1,3-propanediol under anaerobic conditions. Furthermore, the analysis revealed the presence of several non-phenolic organic acids, including citric, succinic, tartaric, and malic acids, which were present in higher concentrations in wild ales compared to standard beer styles. Conversely, GABA, maleic, formic, and fumaric acids were detected in lower concentrations. The absence of pyruvic acid, a compound typically present in beer and wine, suggests either complete microbial conversion by resident microorganisms or degradation during prolonged aging. The amino acid profile of wild ales was found to be diverse, with concentrations that exceeded those typically observed in pale lagers but fell below the levels found in dark beers. This intermediate profile may be indicative of the use of non-roasted or lightly kilned malts, which have been shown to provide moderate amino nitrogen while limiting Maillard reaction products [29].

The saccharide content exhibited marked distinctions as well. Levels of maltodextrin in wild ales were found to be 5 to 14 times lower than those in conventional beers. This is likely due to the occurrence of hydrolysis under acidic conditions (pH 3.2–3.6) or fermentation with *Brettanomyces* spp. [8]. In a similar manner, saccharides such as mannose, xylose, kojibiose, and raffinose were found at reduced levels. Arabinose was an exception, with higher concentrations being observed in certain samples. In contrast, trehalose levels remained consistent across different beer styles. These patterns suggest differential fermentability or degradation of specific saccharides in wild fermentations, potentially shaped by the microbial consortia or adjunct ingredients used [30].

Nucleoside profiling revealed reduced concentrations of adenosine, inosine, and uridine compared to other beer styles. However, guanosine, thymidine, and uracil were present at comparable levels to those found in grape ales and blanche beers [19,27]. Notably, thymine, a rare constituent in beer, was detected in all wild ale samples. The presence of the subject in question has been noted to occur with consistency. This occurrence may be indicative of the turnover of deoxyribonucleic acid (DNA) or ribonucleic acid (RNA) from active microbial populations, or it may be a result of degradation processes that occur during the aging process [31].

The analysis of the core metabolic signatures of wild ales was effectively achieved through the utilization of ^1^H NMR spectroscopy, a technique that facilitates the quantitation of dominant primary metabolites, including organic acids, alcohols, amino acids, saccharides, and other fermentation-derived compounds. These metabolites were identified as being shaped by diverse microorganisms and aging processes. However, to fully capture the metabolic complexity of these beers, LC-MS is indispensable, particularly for detecting and characterizing low-abundance secondary metabolites such as polyphenols, bitter acids, and other trace compounds.

### 2.2. Molecular Composition of Wild Ales via LC-MS

To complement the NMR analysis and broaden the coverage of the metabolome, wild ales were further analyzed via liquid chromatography–mass spectrometry, resulting in the identification of 62 compounds (Table 2).

The spectrum of compounds encompassed by these categories is extensive, including but not limited to a wide variety of organic acids, flavonoids, hydroxybenzoic acids, hydroxycinnamic acids, flavan-3-ols, procyanidins, bitter acids, and prenylated phenols. The significance of this comprehensive range of both primary and secondary metabolites lies in their relevance to the aging process of beer, its flavor profile, and the functional properties that contribute to its overall quality and characteristics. A representative chromatogram is provided in Figure 2.

It is noteworthy that five compounds—succinic, citric, malic, tartaric, and gallic acids—were detected by using both NMR and LC-MS, thus enabling direct cross-platform comparison. This comparison was based on the quantitative NMR profiling results and relative intensity data from the LC-MS analysis. For instance, the application of linear regression analysis to the comparison of NMR and LC-MS measurements of succinic acid (Figure 3) yielded a highly significant relationship (R^2^ = 0.9606), indicating an excellent degree of concordance between the methods. As demonstrated in the Appendix A, analogous regression plots for the remaining shared metabolites are provided.

As expected, phenolic acids and their derivatives were prevalent, with gallic acid, caffeic acid, protocatechuic acid, chlorogenic acid, and ferulic acid-O-hexosides frequently identified in the samples. It is evident that these compounds have their origins in malt, fruit additions, or microbial biotransformation, and that they play key roles in flavor, antioxidant activity, and beer stability. Gallic acid, for instance, was detected in almost all the samples and has been linked to the aging process in wooden barrels or fruit contributions (e.g., grapes, cherries), which is in line with the findings in lambics and oak-aged ales [35].

A significant number of flavonoid glycosides were also identified, including quercetin 3-glucoside, kaempferol 3-glucoside, rutin, and myricetin 3-galactoside. As previously described in the relevant botanical and fruit-enriched beer literature, these flavonols were detected predominantly in samples with dark fruit additions (e.g., W10, W12, W13, W18). This finding aligns with their established occurrence in blackcurrants, cherries, and aronia [37,38,39,40]. Rutin and kaempferol 3-rutinoside, known to be present in grapes and citrus peels, were found to be particularly abundant in Bulgarian and Spanish ales, reflecting their regional ingredient use. Although such flavonol glycosides are widespread in many plants (e.g., tea, buckwheat, pine), their elevated levels in dark fruit-enriched beers result from these fruits’ high concentrations of such compounds [41].

Furthermore, wild ales demonstrated a rich profile of flavan-3-ols and procyanidins, including catechin, epigallocatechin, and procyanidin B3. The presence of these compounds was predominantly observed in ales that incorporated red or purple fruits (e.g., W5, W13, W18). This observation aligns with earlier findings that the incorporation of fruit additions into beer significantly enhances the polyphenol content and antioxidant activity of the final product. The detection of dimeric procyanidins is of particular interest, given the recognized persistence of these compounds through the fermentation and aging processes. This property contributes to their astringency and free radical scavenging capacity [21,42]. Notably, prenylated flavonoids such as xanthohumol, isoxanthohumol, and desmethylxanthohumol were identified in particular in samples brewed with greater hop intensity (e.g., W8, W16, W21). These bitter acids and derivatives are characteristic hop metabolites and have been previously linked to both bitterness perception and biological activity, including estrogenic and antioxidant properties [43,44].

The presence of hop-derived bitter acids, such as iso-α-cohumulone and iso-α-ad/n-humulone, was confirmed in all the samples, thereby reinforcing the essential role of hops even in wild or spontaneous fermentations. These iso-α-acids are formed during the wort boiling and contribute to the core bitterness and microbial stability of beer. Their detection across all the samples—regardless of the fruit or herb additions—confirms that hops remain a consistent component of the formulation in these artisanal styles.

Interestingly, resveratrol, a compound present in red grapes and associated with aging in oak, was detected in Bulgarian and Romanian samples (e.g., W8, W19), thereby corroborating earlier findings on the influence of grape additions or must fermentation in hybrid beer–wine styles [45]. Furthermore, the analysis revealed the presence of additional polyphenols, including rosmarinic acid, ellagic acid, and chlorogenic acid, in select samples. The presence of these compounds may be indicative of the presence of botanical ingredients such as mugwort, pine, or fennel (e.g., W13, W14, W15). However, the concentrations and the detection frequencies of the substances in question were found to be markedly lower than those of the fruit-derived polyphenols. This finding is consistent with the conclusions of previous reports, which indicated that herbal additions typically contribute a narrower and less concentrated phenolic spectrum [46].

In summary, the use of LC-MS profiling has revealed a diverse and complex phenolic signature across the wild ale samples. The data demonstrates the pivotal role of the ingredient type (particularly dark fruits), botanical contributions, barrel aging, and regional brewing practices in determining the chemical and functional properties of wild ales.

### 2.3. Antioxidant Capacity of Wild Ales

The antioxidant capacity of the 22 wild ales, as determined by DPPH and FRAP assays, exhibited significant variation across the samples and countries of origin (Figure 4). The two methods demonstrated a strong correlation (Pearson’s r = 0.914), indicating that the measurement of antioxidant potential through both radical-scavenging (DPPH) and reducing (FRAP) mechanisms was consistent. The DPPH values ranged from 486 to 1970 µmol TE/mL (average: 1074), while the FRAP values ranged from 2.24 to 9.12 µmol/mL beer (average: 4.34).

The sample exhibiting the highest degree of antioxidant activity in both assays was W13, which was brewed with blackcurrant fruit and fennel seed. In contrast, the lowest DPPH activity was observed in W11 (a combination of rhubarb and blackcurrant leaf), while the lowest FRAP value was found in W14, which contained mugwort. This substantial discrepancy underscores the predominant impact of the fruit type and composition on the antioxidant potential. The present finding is corroborated by the extant literature: berry fruits typically contain 2.9 to 5.9 times more total phenolics and up to 5.8 times higher antioxidant activity than corresponding leaves [47]. As demonstrated in the relevant literature, blackcurrant leaves do contain valuable phenolics, such as quercetin and myricetin derivatives. However, they lack the high anthocyanin concentrations characteristic of the fruit, which have been shown to be potent contributors to antioxidant capacity. In the context of brewing, the efficiency of phenolic extraction is also a contributing factor. However, under the assumption of analogous extraction kinetics, the elevated intrinsic levels of anthocyanins and other polyphenols in dark fruits are likely to result in enhanced antioxidant activity in the final product. The majority of wild ales enriched with dark-colored fruits—including strawberries (W1), gooseberries (W2), raspberries (W3, W18, W21, W22), blackcurrants (W13, W17), cherries (W7, W17), red grapes (W3, W8, W21), blueberries, and plums (W5) have shown a consistent tendency to exhibit the highest antioxidant values. These findings are consistent with previous research that demonstrated higher antioxidant activity in fermented beverages made with berries or red grapes compared to those made with lighter fruits. Ljevar et al. found that blackberry, cherry, raspberry, and blackcurrant wines contained the highest total phenolic content and antioxidant capacity among 32 Croatian fruit wines, significantly exceeding the levels found in apple wines [48]. Dark fruits are recognized for their richness in flavonoids, phenolic acids, and anthocyanins, compounds that have been shown to possess free radical-scavenging and metal-reducing activity. Analogous trends have been observed in wild ales. Cherry- and raspberry-based lambics have been shown to exhibit considerably higher phenolic content and antioxidant activity in comparison to conventional pale ales, lagers, and wheat beers. Correspondingly, wild ales enriched with dark-colored fruits exhibited levels that consistently surpassed the conventional antioxidant thresholds documented for conventional beer varieties, which typically range below ~1.0 µmol TE/mL in DPPH assays. In comparison with the reference values of other beer types, the wild ales in this study exhibited DPPH values that were lower than those of bilberry beers (2089–2465 µmol TE/mL) but analogous to those reported for sour ales (897–996 µmol TE/mL) and, in general, higher than lagers (240–1350 µmol TE/mL) [21,42]. This finding aligns with earlier observations that the incorporation of fruit significantly enhances the antioxidant activity in beers [35]. The addition of botanical and herbal elements, including mugwort (W14), pine bark (W16), and fennel seed (W15), was found to be associated with antioxidant activity levels that were moderate to low. Despite the extensive documentation of the bioactivity of these botanicals, their contribution to the overall antioxidant capacity of beer appears to be negligible. The underlying cause of this phenomenon may be attributable to a diminution in the extraction efficiency, a reduction in dosage, or the degradation of active phenolic constituents over time, particularly under conditions of oxidation during the fermentation and aging processes. The present findings are consistent with earlier research that identified the presence of unique phenolics (e.g., hydroxycinnamic acids, flavonol glycosides) in herbs, yet their concentrations and solubility are frequently inadequate to exert a substantial effect on antioxidant assays within the beer matrix [46]. Nevertheless, the present study revealed that the addition of herbal ingredients did not yield antioxidant levels comparable to those achieved by dark fruit additions, emphasizing the pivotal role of ingredient selection in optimizing the enhancement of the functional potential.

The effect of barrel aging on the antioxidant potential of the wine was found to be modest, particularly when fruit additions were employed in conjunction with the aging process (e.g., W13, W17, W22). However, the findings indicated that the process of aging alone was not a reliable predictor of the antioxidant strength. In the context of beer aging, the influence of the ingredient composition on the antioxidant values has been demonstrated to be more dominant than that of the aging duration or method. This was evidenced by the lower antioxidant values exhibited by beers aged in oak barrels without fruit (e.g., W10, W11, W14). No statistically significant correlation was observed between the antioxidant activity and aging time; both short-aged (≤18 months) and long-aged (>18 months) beers spanned the full range of antioxidant values. These findings are further supported by the ANOVA results, which revealed significant differences in the antioxidant activity between the ingredient groups for both DPPH (*p* = 0.018) and FRAP (*p* = 0.015). However, no significant differences were observed based on the aging duration (*p* > 0.5).

The findings of this study suggest that the antioxidant capacity of wild ales is predominantly influenced by the type of fruit utilized, particularly those that are phenol-rich and dark in color. The duration of aging and the method of aging appear to moderate this effect to a lesser extent, but do not independently determine the antioxidant potential under the studied conditions.

### 2.4. Impact of Fruit and Herb Additions on the Molecular Composition of Wild Beers

A key factor contributing to the molecular complexity of wild beers is the diversity of the raw ingredients, particularly the incorporation of fruits and herbs. To elucidate the influence of these ingredients on the chemical composition, multivariate analyses were performed using integrated NMR, LC-MS, and spectrophotometric datasets. The results obtained provide a comprehensive perspective on the impact of the ingredient selection on the metabolomic profile of wild ales.

To assess the influence of the added fruits and herbs on the molecular composition of wild ales, a hierarchical clustering analysis was performed using combined NMR, LC-MS, and spectrophotometric data. The resulting dendrogram (Figure 5) revealed three primary clusters corresponding to the type of ingredients added: (i) wild ales with dark fruits (*n* = 13), (ii) ales with light fruits (*n* = 4), and (iii) those without added fruits (*n* = 5). The dark fruit cluster included samples enriched with ingredients such as strawberries, gooseberries, blueberries, red and black grapes, raspberries, blackcurrants, plums, aronia, and cherries (W1–W3, W5, W7, W8, W13, W15, W17–W19, W21, W22). The light fruit cluster (W4, W9, W16, W20) was distinguished by the incorporation of citrus fruits, peaches, white grapes, mandarins, oranges, and limes. The third cluster comprised beers brewed without fruit additions (W6, W10–W12, W14), instead incorporating herbal or cereal ingredients such as blackcurrant leaves, mugwort, spruce, blackthorn, pine, birch bark, and spelt.

A notable observation was the compositional similarity exhibited by beers containing light fruits and those brewed without fruit additions, as compared to the dark fruit group. This observation may be partially attributed to the overlap in their profiles of saccharides and glycosylated phenolics, or to the relatively mild phenolic and pigment content in light fruits compared to darker ones. It should be noted that W9, brewed with white grapes, was found to be somewhat distinct within the light fruit group, a distinction that may be attributable to the specific phenolic and aromatic contributions of *Vitis vinifera* cultivars.

Subsequent substructure analysis within the dark fruit cluster yielded the identification of four subclusters: (i) W2, W3, W13, W17; (ii) W5, W7; (iii) W8, W19; and (iv) W1, W15, W18, W21, W22. The hierarchical substructure analysis of the “dark fruit” cluster revealed four distinct subclusters, each of which reflected unique biochemical signatures imparted by the specific fruit combinations and fermentation dynamics. Subcluster 1 (W2, W3, W13, W17) was characterized by high acidity in fruits, notably gooseberries and blackcurrants and a notable diversity of anthocyanin. Significant levels of organic acids such as citric and malic acids were also observed. The grouping of samples indicates the presence of a shared underlying profile of tartness and intense red/dark fruit pigmentation. Subcluster 2 (W5, W7) included beers with blueberries and plums (W5), and Picota cherries (W7). In comparison with the preceding subcluster, these fruits frequently exhibit a less pronounced tartness, and occasionally a sweeter profile, while maintaining a high concentration of various anthocyanins. Subcluster 3 (W8, W19) exclusively featured grape-derived beers (Pinot Noir, Fragola), thereby underscoring the outsized influence of grape-specific metabolites on their clustering. The homogeneity observed in this study indicates that the geographical origin of the grapes may have a more significant impact on the metabolic profiles than other factors such as botanical additives. Subcluster 4 (W1, W15, W18, W21, W22) demonstrated the highest level of compositional diversity, incorporating strawberries, raspberries, and aronia with non-fruit adjuncts (buckwheat, lavender). Despite this variability, the presence of shared raspberry content across three samples (W18, W21, W22) and the identification of the Cabernet Sauvignon grape (W21) may serve to anchor their grouping by common flavonols [40].

To further explore and validate the chemical differences among the three major ingredient-based groups, a supervised OPLS-DA was performed using 112 variables. The classes were defined as follows: dark fruit (*n* = 13), light fruit (*n* = 4), and no fruit (*n* = 5). In order to ascertain the most influential metabolites in group discrimination, Variable Importance in Projection (VIP) values were calculated. The final sample comprised a total of 23 variables with VIP scores greater than 0.85 selected for model optimization, including three alcohols (2-phenylethanol, ethanol and methanol), five saccharides (kojibiose, maltodextrin, mannose, αα-trehalose and xylose), two nucleobases (thymine and uracil), five organic acids (succinic acid, 4-hydroxybenzoic acid, protocatechuic acid, dihydroxybenzoic acid and 4-hydroxyphenylacetic acid), six flavonoids (naringenin, isoxanthohumol, kaempferol 3-glucoside, quercetin 3-glucoside, myricetin 3-galactosideand hesperidin), and two antioxidant indicators (DPPH and FRAP). The resulting OPLS-DA model (2 predictive + 3 orthogonal components) exhibited adequate fit parameters (R^2^X = 0.698, R^2^Y = 0.816), though its predictive ability was only moderate (Q^2^ = 0.153). The performance of the model was evaluated through the analysis of the ROC curve and Area Under the Curve (Appendix A). The classification accuracy of the model was determined by means of a misclassification table (Appendix A).

As illustrated in Figure 6, the OPLS-DA score plot corroborated the clustering results, thereby clearly delineating the separation of the three groups. A study of wild ales with light fruits revealed higher levels of maltodextrin, xylose, mannose, kojibiose, ethanol, methanol, and citrus-specific flavonoids, including naringenin and hesperidin. The aforementioned compounds have been shown to be consistent with the polyphenol content that is typically found in citrus products [49]. W9 has been found to contain elevated levels of 2-phenylethanol, a compound that has been associated with floral, rose-like aromas and that is characteristic of grape-derived metabolites [50]. The elevated levels of ethanol and methanol may be indicative of the disparities in fruit fermentability or pectin breakdown during secondary fermentation.

The group of beers devoid of fruit additions exhibited heightened levels of kaempferol-3-glucoside, quercetin-3-glucoside, myricetin-3-galactoside, thymine, and uracil. These compounds are likely derived from added herbs (e.g., mugwort, pine, spruce) and cereals (e.g., spelt, rye), which are known sources of glycosylated flavonols and nucleobases, especially in wild fermentation contexts [51]. Specifically, W10 and W12 were found to contain elevated concentrations of kaempferol-3-glucoside and quercetin-3-glucoside, thereby supporting the hypothesis that the inclusion of bark, pine, and herbaceous components contributes to this distinctive phytochemical fingerprint. In contrast, a heightened antioxidant activity and elevated concentrations of succinic and protocatechuic acids were observed in dark fruit beers. The presence of these compounds has been observed in fruit-rich fermentations, and they have been associated with an increased sensory complexity, preservation potential, and health-related bioactivity [39,41].

These findings emphasize the significance of the ingredient selection in shaping the chemical profiles of wild beers and encourage further exploration of how other factors, such as barrel aging, contribute to chemical transformations over time.

### 2.5. Molecular Changes Induced by the Long-Term Aging of Wild Beers

It is evident that, whilst the raw ingredients play a significant role in the composition of wild beer, time is a critical factor in the shaping of the final product. The process of barrel aging has been demonstrated to induce chemical transformations through the agency of ongoing microbial activity, wood–beer interactions, and oxidative processes. In order to examine the metabolic evolution associated with the process of aging, the samples were stratified according to the maturation time.

In order to investigate the chemical transformations associated with extended barrel aging and assess the potential authenticity markers, wild ales were stratified into two temporal groups: those aged up to 18 months (*n* = 9) and those aged for more than 18 months (*n* = 13). The construction of an OPLS-DA model was then based on 22 discriminant compounds with VIP > 0.88. The resulting model (one predictive + three orthogonal components) exhibited robust statistical performance (R^2^X = 0.604, R^2^Y = 0.901, Q^2^ = 0.591). The score plot (Figure 7) demonstrated a clear separation between the two aging classes.

A study of wild ales aged for a period exceeding 18 months revealed elevated levels of several amino acids, including leucine, valine, and alanine. Additionally, volatile aldehydes and ketones (acetaldehyde, ethyl acetate, and acetoin), organic acids (acetic acid, gallic acid, tartaric acid), and other fermentation-derived metabolites (1,3-propanediol, choline, and arabinose) were identified. In contrast, beers that underwent a shorter aging period exhibited increased concentrations of organic acids (citric, malic), betaine, phenolic compounds (4-vinylguaiacol, catechin, procyanidin B3), and ferulic acid-O-hexosides. Notably, despite these compositional differences, DPPH and FRAP assays demonstrated no significant variation in the total antioxidant capacity between the various age groups. The results of the present study are consistent with those of the previous research conducted on barrel-aged and spontaneously fermented beers. These earlier studies demonstrated a gradual accumulation of amino acids and volatile compounds over time resulting from proteolysis, microbial metabolism, and wood–beer interactions [30]. In particular, an increase in the levels of branched-chain amino acids (BCAAs), such as leucine and valine, has been demonstrated to be associated with an extended fermentation process by using *Brettanomyces* and *Pediococcus* species. These species have the capacity to hydrolyze the proteins and peptides that are released during the process of yeast autolysis or malt residues [52,53]. These amino acids contribute to the nutritional complexity and serve as precursors for the higher alcohols and esters involved in flavor development.

The presence of elevated levels of acetaldehyde, ethyl acetate, and acetoin in the older samples indicates heightened oxidative and microbial activity. Acetaldehyde, a known aging marker in both wine and beer, can accumulate through ethanol oxidation or the metabolism of pyruvate by acetic acid bacteria and *Brettanomyces* [52]. Ethyl acetate is frequently the most prevalent ester in aged sour beers, contributing to fruity and solvent-like aromas, particularly at elevated concentrations. Acetoin, which is derived from the reduction of diacetyl or via the butanediol pathway, has been reported to increase during the maturation of sour and mixed fermentation beers. This increase may contribute to buttery or creamy notes. The elevated presence of gallic acid in the older wild ales may indicate progressive degradation of tannin-rich precursors and lignin-like compounds extracted from oak barrels [54,55,56]. This particular phenolic acid is also recognized as a by-product of microbial hydrolysis and deconjugation polyphenols that are derived from plants. These processes are enhanced by prolonged exposure to *Brettanomyces* and other microbes that are resident in the barrel [6]. In a similar manner, the accumulation of 1,3-propanediol, choline, and arabinose is probably the result of the continued fermentation of residual or barrel-derived carbohydrates by slow-acting fermentative and anaerobic bacteria (e.g., *Lactobacillus*, *Leuconostoc*) [54]. In contrast, beers that had been aged for up to 18 months exhibited higher concentrations of labile compounds, including citric acid and malic acid. It is evident that these compounds are the primary fermentation products, and it is well established that their levels typically decrease with prolonged aging due to microbial conversion processes such as malolactic and citric acid metabolism. The presence of higher amounts of betaine, ferulic acid-O-hexosides, catechin, and procyanidin B3 in younger beers suggests that there has been a lesser degree of degradation or transformation of phenolics derived from cereal and fruit. These compounds have been demonstrated to be susceptible to several chemical processes. They have been shown to be vulnerable to oxidation, microbial hydrolysis, or polymerization during extended aging. It is further suggested that these substances may be capable of serving as indicators of shorter maturation periods [57]. For instance, ferulic acid derivatives are prevalent in both wort and fruit additions and are recognized precursors for *Brettanomyces*-mediated volatile phenol production (e.g., 4-vinylguaiacol). However, a decline in these derivatives was observed in longer-aged samples, which may be attributed to further transformation into 4-ethylguaiacol [52].

The findings, when considered collectively, demonstrate that barrel aging beyond 18 months is associated with advanced chemical evolution and fermentation-derived complexity. This supports the use of barrel aging not only for flavor development but also as a potential authenticity marker. The metabolic signatures that have been observed could be used in future studies to verify claims regarding the aging process in artisanal beer production, or to trace the maturation of beer in the absence of controlled storage records.

This chemical variation, associated with the aging process, provides a framework for investigating the possible effect of the production region on the characteristics of wild ale. This is particularly relevant given the potential impact of regional practices on factors such as aging duration, ingredient selection, and microbial consortia composition.

### 2.6. Regional and National Influences on the Molecular Composition of Wild Ales

In consideration of the heterogeneity that characterizes traditional brewing practices throughout Europe, it can be hypothesized that the geographic origin may exert a substantial influence on the final composition of wild ales. In order to ascertain the existence of regional trends, a classification system was devised to categorize beers based on their country of production. These beers were then subjected to a thorough analysis to detect and identify molecular markers unique to specific countries.

To explore the potential national styles and flavor signatures, wild ales were classified based on their country of production. The countries included in the study were Poland (*n* = 4), Spain (*n* = 3), Bulgaria (*n* = 3), the Netherlands (*n* = 7), Romania (*n* = 2), and Denmark (*n* = 3). A model based on OPLS-DA and 40 discriminatory variables (VIP > 0.9) exhibited adequate explanatory power (5 + 2 components; R^2^X = 0.795, R^2^Y = 0.839, Q^2^ = 0.544), with the score plot distinctly differentiating all the groups (Figure 8).

The biplot interpretation revealed several country-specific chemical trends. Polish wild ales were found to contain betaine, glycerol, xylose, arabinose, catechin, procyanidin B3, and ferulic acid-O-hexoside. This profile indicates a marked emphasis on specific malt bills, which may encompass unmalted grains, in conjunction with conventional brewing methodologies that facilitate the conservation of phenolics derived from cereals. The observed phenolic profile is indicative of either constrained microbial activity or reduced aging duration, which limits the typical transformation of these compounds [58].

The analysis of Spanish ales revealed the presence of rutin and kaempferol 3-rutinoside, flavonoid glycosides that are commonly found in various fruits and other plant sources. The compounds in question may be derived from spontaneous fermentations incorporating fruits or other plant-based adjuncts [37,40]. This is indicative of regional preferences for bitter–fruity profiles in beers.

Notably, the Bulgarian samples exhibited elevated concentrations of leucine, valine, tartaric acid, resveratrol, various alcohols (1-propanol, isopentanol), 1,3-propanediol, and acetaldehyde. The presence of tartaric acid and resveratrol, which are typically markers associated with grapes and red wine, supports the likelihood of grape or must addition during the brewing process. This finding is consistent with recent regional experiments in co-fermenting beer and wine substrates [45]. The elevated levels of amino acids and alcohols indicate the presence of active mixed fermentation, involving *Saccharomyces*, *Brettanomyces*, and lactic acid bacteria (LAB) [54].

Dutch wild ales exhibited a notable abundance of thymine alongside key hop-derived compounds, including apigenin, isoxanthohumol, and iso-α-acids (iso-α-cohumulone, iso-α-ad/n-humulone). The presence of these specific hop compounds, particularly the bittering iso-α-acids and the unique prenylflavonoid isoxanthohumol, is highly indicative of a pronounced hop-forward profile. Isoxanthohumol and its precursors are typically derived from aged hops or can be significantly extracted through specialized hopping techniques, such as dry hopping and whirlpool hopping [44]. Consequently, these robust hop profiles are likely indicative of a national preference for highly hopped, bitter beers.

A thorough investigation into the chemical composition of Romanian beers was undertaken, resulting in the identification of a plethora of compounds, including amino acids, glycerol, arabinose, lactose, and 5-hydroxymethylfurfural (5-HMF). The significant presence of lactose is indicative of the utilization of milk sugars, thereby imparting unfermentable sweetness and body to the beer. Elevated levels of histidine and 5-HMF are indicative of Maillard reaction products, suggesting that specific boiling practices or the addition of roasted malt may be contributing factors [59]. This chemical profile is unique in its indication of a preference for richer, sweeter, and potentially malty or darker beer characteristics among the Romanian samples.

A study of Danish wild ales revealed elevated concentrations of alanine, succinic acid, methanol, a variety of sugars (e.g., kojibiose, mannose, and trehalose), and 4-hydroxybenzoic acid, a compound with mild antimicrobial properties that is often derived from lignin breakdown. The sugar profile may be indicative of the use of adjuncts such as rice, corn, or novel saccharification strategies, while the presence of methanol could be attributed to pectin-rich fruit additions undergoing extensive fermentation [60].

The results of this study indicate that, despite the utilization of wild or spontaneous fermentation across all the groups, region-specific ingredients, cultural preferences, and brewing techniques imprint unique metabolomic signatures. The presence of grape-derived metabolites in the Bulgarian samples suggests the possibility of terroir-driven experimentation. Distinguishing characteristics such as the hop-centric profile characteristic of Dutch ales or the compounds contributing to unique body and mouthfeel profiles present in Polish and Romanian beers, serve to reflect the distinct brewing identity or stylistic focus of these regions.

The present study encompasses wild ales originating from multiple countries (namely Poland, Spain, Bulgaria, the Netherlands, Romania, Denmark). However, it should be noted that the sample sizes per country are uneven and relatively small. This limitation precludes a comprehensive evaluation of the impact of the geographical origin or local brewing practices on the antioxidant activity. It is acknowledged that regional variations in raw materials, microbial communities, and production methods may contribute to compositional differences that have not been fully addressed in this study. It is recommended that future studies employ larger, more balanced datasets and controlled sampling across regions in order to clarify the extent to which geographic factors affect the chemical profiles of wild ales.

### 2.7. Correlation Analysis: Connecting Molecular Composition to Antioxidant Activity

In order to elucidate the molecular contributors to the antioxidant activity in wild ales, Pearson correlation analysis was performed between antioxidant assays (DPPH and FRAP), and compounds were identified via NMR and LC-MS. A moderate positive correlation (r > 0.4) was observed for several metabolites, including 2,3-butanediol, succinic acid, malic acid, methanol, αα-trehalose, and key phenolic compounds such as 4-hydroxybenzoic acid, dihydroxybenzoic acid, catechin, ellagic acid, epigallocatechin, gallocatechin, and procyanidins (Figure 9). The strongest correlations were found for salicylic acid (r = 0.72, DPPH), indicating that it is the primary contributor to radical scavenging or redox capacity in the wild beer matrix.

These compounds are derived primarily from fruit or herb additions and microbial bioconversion during the aging process. Phenolic acids and flavonoids have been demonstrated to exert antioxidant effects through mechanisms including electron or hydrogen donation, radical stabilization, and metal chelation. Furthermore, it has been demonstrated that organic acids are capable of modulating the redox potential either indirectly or by acting as markers of microbial metabolism [61]. The correlation of αα-trehalose and 2,3-butanediol, both of which have been associated with yeast and LAB stress response and osmoprotection, further implicates the microbial contributions to antioxidant preservation [62].

In contrast, moderate negative correlations were identified with 2-phenylethanol and trigonelline. These compounds have been frequently linked to oxidative stress. For instance, 2-phenylethanol, a yeast-derived aromatic alcohol, has been observed to increase during the aging process or thermal exposure, which may be indicative of antioxidant depletion [50].

The correlation trends indicate that the antioxidant potential of wild ales is modulated by two main factors: the phenolic content, which is largely derived from adjunct ingredients, and specific microbial metabolites. Furthermore, an inverse correlation between stress-related compounds and redox capacity has been observed.

### 2.8. Limitations and Future Perspectives

This study provides the most comprehensive metabolomic characterization of wild ales to date. It integrates NMR, LC-MS, and antioxidant assays across 22 samples from six countries. While it offers new insights into ingredient- and region-specific signatures, several limitations, including those detailed in Section 2.6, merit consideration. Although the sample size is diverse and covers key brewing regions, it may not capture the full variability of global production. Additionally, differences in spontaneous fermentation (e.g., microbial community variability) and barrel-aging conditions (e.g., wood type and reuse history) between batches could affect the reproducibility. Our approach surpasses previous GC-MS-focused beer metabolomics studies by capturing both volatile and non-volatile compounds and linking them to functional antioxidant properties. Unlike earlier research emphasizing microbial succession, we relate chemical profiles to specific production variables, such as fruit/herb additions, aging duration, and geography, to support the authenticity assessment. Identifying biomarkers associated with ingredient use and aging duration provides practical tools for quality control. These advances lay the groundwork for future research on scaling up to larger datasets and integrating adjunct dosing control, microbial–metabolite network analysis, and sensory data on consumer preferences. The potential role of hop-derived compounds in shaping the microbial stability and fermentation dynamics is also an important area for future investigation.

## 3. Materials and Methods

### 3.1. Materials

For the compound identification, a range of authenticated analytical standards was used. The following standards were obtained from PhytoLab GmbH & Co. KG (Vestenbergsgreuth, Germany): quinic acid (≥98%, CAS: 77-95-2), gallic acid (≥95%, CAS: 149-91-7), (+)-catechin (≥98.0%, CAS: 154-23-4), ellagic acid (≥95.0%, CAS: 476-66-4), hyperoside (≥95.0%, CAS: 482-36-0), guajaverin (≥95.0%, CAS: 22255-13-6), trans-p-coumaric acid (≥99%, CAS: 501-98-4), caffeic acid (≥98%, CAS: 501-16-6), apigenin (≥95%, CAS: 520-36-5), naringenin (≥95%, CAS: 67604-48-2), kaempferol (≥95%, CAS: 520-18-3), luteolin (≥95%, CAS: 491-70-3), chrysoeriol (≥90%, CAS: 491-71-4), quercetin (≥95%, CAS: 117-39-5), (−)-epigallocatechin (≥95%, CAS: 970-74-1), caftaric acid (≥95%, CAS: 117-39-5), isorhamnetin (≥95%, CAS: 480-19-3), neochlorogenic acid (≥95%, CAS: 906-33-2), rosmarinic acid (≥98%, CAS: 20283-92-5), orientin (≥98%, CAS: 28608-75-5), kaempferol 3-glucoside (≥95%, CAS: 480-10-4), quercetin 3-glucoside (≥95%, CAS: 482-35-9), myricetin 3-galactoside (≥98%, CAS: 15648-86-9), 3,5-dicaffeoylquinic acid (≥95.0%, CAS: 2450-53-5), 4,5-dicaffeoylquinic acid (≥95.0%, CAS: 14534-61-3), procyanidin B3 (≥95%, CAS: 23567-23-9), kaempferol 3-rutinoside (≥95%, CAS: 17650-84-9), rutin (≥95%, CAS: 153-18-4), quercetin 3-sophoroside (≥95%, CAS: 18609-17-1), isoxanthohumol (≥95%, CAS: 521-48-2), xanthohumol (≥98%, CAS: 6754-58-1). Additional standards were purchased from Sigma-Aldrich (Darmstadt, Germany): chlorogenic acid (≥95%, CAS: 327-97-9), 4-hydroxybenzoic acid (≥99%, CAS: 99-96-7), and protocatechuic acid (≥99%, CAS: 99-50-3). Other reagents included L-aspartic acid (≥99%, CAS: 56-84-8, Fluka Chemie AG, Switzerland), L-glutamic acid (analytical grade, CAS: 56-86-0, Reanal, Budapest, Hungary), and salicylic acid (analytical grade, CAS: 69-72-7, Chim-Spectar Ltd., Sofia, Bulgaria). LC-MS-grade acetonitrile and methanol (Chromasolv^®^) were obtained from Honeywell Riedel-de Haën (Seelze, Germany). Formic acid (LC-MS grade Lichropur™, 97.5–98.5% purity, CAS: 64-18-6) was sourced from Sigma-Aldrich (Buchs, Switzerland). Ultrapure water (resistivity ≤ 0.055 µS/cm) was produced using a Smart2Pure 12 UV/UF system (Thermo Electron LED GmbH, Langenselbold, Germany).

For the antioxidant activity assays, various chemicals were employed. 2,2-Diphenyl-1-picrylhydrazyl (DPPH, 95%) was obtained from Sigma-Aldrich (Germany). Iron (II) sulfate heptahydrate (pure for analysis) and absolute ethanol (99.9%, pure for analysis) were sourced from Valerus (Sofia, Bulgaria). Iron (III) chloride (98% pure, anhydrous) and 2,4,6-tri(2-pyridyl)-1,3,5-triazine (99%) were purchased from Acros Organics (Germany and Austria, respectively). Sodium acetate trihydrate (pure for analysis) came from Chim-spectar (Bulgaria). Acetic acid (puriss. p.a., ≥99.8%) and hydrochloric acid (puriss. p.a., ≥37%) were supplied by Honeywell (Seelze, Germany and Esch, Austria, respectively), while HPLC-grade methanol (≥99.8%) was obtained from Fisher Scientific (Loughborough, UK).

### 3.2. Beer Samples

A total of 22 wild ales from six European countries were purchased from three specialized craft beer shops in Bulgaria. The selection criteria were centered on commercial availability and the labeling as wild ales (without distinguishing substyles) with a view to creating a general, representative chemical profile of this beer category. Appendix A provides a comprehensive list of the beer names, breweries, countries of origin, and sample abbreviations.

### 3.3. Sample Preparation

All the beer samples were degassed to remove carbonation using ultrasonication. For the NMR analysis and antioxidant assays, degassing was performed for 15 min in an ultrasonic bath. For the LC-MS analysis, the samples were degassed for 30 min to ensure complete CO_2_ removal, which can interfere with ionization and chromatography.

For the NMR analysis, 500 μL of degassed beer was mixed with 50 μL of deuterated phosphate buffer solution (pH 4.4) containing 0.1% TSP (3-(trimethylsilyl)-2,2,3,3-tetradeuteropropionic acid sodium salt) as the internal reference, 0.05% NaN_3_ as a preservative, and D_2_O for field-frequency locking. The mixture was vortexed briefly and transferred into 5 mm NMR tubes for measurement.

For the LC-MS analysis, the degassed samples were diluted 1:1 (*v*/*v*) with ultrapure water and filtered through 0.22 μm hydrophilic PTFE syringe filters (13 mm diameter) to remove particulates and colloidal matter. The filtered samples were subsequently transferred to vials prior to injection.

To perform the antioxidant assays, the beer samples were degassed and diluted with distilled water to create working solutions. A 5-fold dilution was used for the DPPH assay and a 10-fold dilution was used for the FRAP assay.

### 3.4. NMR Analysis

Proton nuclear magnetic resonance (^1^H NMR) spectra were acquired on a Bruker Avance NEO 400 MHz spectrometer equipped with a broadband-observe (BBO) probe and maintained at 300.0 ± 0.1 K. Water suppression was achieved using the zgcppr pulse sequence to reduce solvent signal intensity. Acquisition parameters were set as follows: spectral width of 13.2 ppm, 42,104 time-domain points (TD), acquisition time (*aq*) of 4.0 s, relaxation delay (d1) of 4.0 s, 256 transients, and 16 dummy scans to ensure steady-state magnetization. The resulting free induction decays (FIDs) were processed with exponential multiplication applying 0.3 Hz line broadening before Fourier transformation. Phase and baseline corrections were performed manually in MestreNova 14.2.3. The internal standard signal of TSP was calibrated at 0 ppm and used for chemical shift referencing and quantitative calculations.

Metabolite identification was performed by comparing chemical shifts and coupling patterns to entries in established reference databases—HMDB and BMRB—as well as to previously published assignments. To support identification and resolve overlapping peaks, 2D experiments, including HSQC and TOCSY, were conducted on representative samples. Additional validation was performed for the key metabolites, such as amino acids, organic acids, and sugars, by spiking authentic standards into the beer samples to confirm the signal assignments. Due to the complexity of the beer matrix and frequent spectral overlap among the compounds, deconvolution techniques (line fitting) were applied to selected regions of the spectrum to accurately quantify signals that could not be directly integrated. The quantification of metabolites with confident assignments was carried out using a standardized calculation method consistent with protocols approved by the International Organisation of Vine and Wine (OIV). This method enables reproducible and comparable concentration determinations across samples [63].

### 3.5. HPLC-MS/MS

The HPLC-MS/MS analysis was carried out using a Q Exactive Plus^®^ hybrid quadrupole-Orbitrap^®^ mass spectrometer (HRMS/MS) equipped with a heated electrospray ionization (HESI) source, coupled with a Vanquish UHPLC system (Thermo Fisher Scientific, Bremen, Germany). HPLC analyses were conducted using a Restek Raptor C18 column (2.7 µm, 100 × 2.1 mm) maintained at 40 °C, equipped with a direct-connection column, 2.1 mm (Thermo Fisher Scientific, Bremen, Germany). Gradient elution was applied at a flow rate of 0.3 mL/min, with an injection volume of 5 µL. The mobile phase consisted of 0.1% (*v*/*v*) formic acid in water (eluent A) and acetonitrile (eluent B). The following gradient was employed: 0–4 min, 2–25% B; 4–7 min, 25–75% B; 7–8 min, 75–95% B; 8–10 min, 95% B; 11–14 min, 2% B. Mass spectrometric detection was performed using an HESI source under the following conditions: spray voltage −2.90 kV, capillary temperature 320 °C, sheath gas at 30 arbitrary units, auxiliary gas at 6 arbitrary units, sweep gas 0, and S-Lens RF level set to 50 V. Nitrogen served both as a nebulizer and collision gas (HCD). Full-scan MS spectra were acquired in the negative ion mode at a 70,000 resolution, AGC target of 1e6, a maximum injection time (IT) of 80 ms, and a scan range of 100 to 1200 *m*/*z*. Data-dependent MS^2^ experiments were carried out with a resolution of 35,000, AGC target of 1e5, maximum IT 50 ms, isolation window of 2.0 *m*/*z*, and stepped NCE of 20, 40, and 70. Data acquisition, processing, and compound identification were conducted using Xcalibur version 4.2 SP1, FreeStyle version 1.5, and Compound Discoverer version 3.3 SP3 (Thermo Fisher Scientific, Bremen, Germany). This untargeted LC-MS approach enabled qualitative or semi-quantitative identifications of compounds without determining their absolute concentrations. The presence and absence of the identified 62 compounds in the studied samples are presented in Appendix A.

### 3.6. Antioxidant Activity Assays

The antioxidant potential of the wild ales was evaluated using DPPH radical scavenging and FRAP (ferric-reducing antioxidant power) assays.

For the DPPH assay, a modified method based on the work of Nenadis and Tsimidou was employed [22]. Diluted samples (100 µL) were mixed with 2 mL of a 0.1 mM DPPH solution in methanol. The mixture was vortexed and then centrifuged at 13,000 rpm for 10 min (Sigma 1–14, Osterode am Harz, Germany) to remove particulates. The resulting supernatant was incubated in the dark at room temperature for 30 min to prevent photodegradation. Absorbance was measured at 517 nm using a Thermo Scientific Helios Gamma UV–Vis spectrophotometer. The results were expressed as µmol Trolox equivalents per mL (µmol TE/mL) using a Trolox standard curve.

The FRAP assay was adapted from Benzie and Strain and Tomasina et al. [23,64]. The FRAP reagent was freshly prepared by mixing acetate buffer (pH 3.6), TPTZ (2,4,6-tri(2-pyridyl)-1,3,5-triazine) solution in HCl, and FeCl_3_·6H_2_O. For each test, 100 µL of diluted beer was combined with 3 mL of FRAP reagent and incubated in the dark at room temperature for 30 min. The absorbance was measured at 593 nm against a reagent blank. The results were expressed as µmol Fe^2+^ equivalents per mL, calculated from an FeSO_4_ standard curve.

All the measurements were performed in triplicate, and average values were used for subsequent statistical analysis.

### 3.7. Statistical Analysis

All the quantitative results from the NMR profiling, the relative intensity data from the LC-MS analysis, and the antioxidant assay measurements were standardized (z-score transformation) prior to statistical modeling to ensure the comparability across the variables with different scales and units. Orthogonal Partial Least Squares-Discriminant Analysis (OPLS-DA) was applied using SIMCA 18.0 software (Umetrics, Umeå, Sweden) to evaluate the group differences and identify discriminating metabolites. The class definitions included the ingredient additions (e.g., fruit type), aging duration, and country of production. The model quality was assessed using explained variance metrics (R^2^X and R^2^Y) and predictive ability (Q^2^). To validate the classification performance, 7-fold cross-validation was employed, while Receiver Operating Characteristic (ROC) curves and their respective Area Under the Curve (AUC) values (Appendix A), alongside misclassification tables (Appendix A) provided further assessment of the sensitivity and specificity. To facilitate the interpretation of the OPLS-DA results, loading plots (Appendix A) and VIP score plots (Appendix A) were generated in order to visualize the variables that contribute most strongly to class separation. The loadings plots illustrate the contribution of each metabolite to the class separation along the predictive component, while variables with higher VIP scores are considered stronger contributors to discrimination.

Hierarchical clustering analysis (HCA) was used to provide unsupervised insights into the data structure. By applying Ward’s linkage method, which minimizes within-cluster variance, HCA visualized natural, compact, and interpretable sample groupings based on the metabolomics data.

A one-way ANOVA was performed in Microsoft Excel to assess the statistical significance of the differences in the antioxidant capacity (DPPH and FRAP values) across the ingredient groups and aging duration categories.

Pearson correlation analysis was performed in PAST 4.10 software [65] to evaluate the relationships between the antioxidant capacity (DPPH and FRAP values) and individual metabolite abundances identified by using both NMR and LC-MS. The correlation significance was assessed using calculated *p*-values, ensuring that only statistically meaningful associations were interpreted. This approach aimed to identify the specific compounds that contribute to antioxidant activity and facilitate the discovery of potential antioxidant markers in the complex beer matrix.

## 4. Conclusions

This study provides an in-depth characterization of wild ales through integrated nuclear magnetic resonance and liquid chromatography–mass spectrometry metabolomics, revealing how diverse production variables shape their molecular profiles. The integration of fruit and herb components yielded the introduction of a range of polyphenols, saccharides, and organic acids. A substantial proportion of these compounds are retained or transformed during the aging process. Prolonged maturation has been shown to be associated with elevated levels of sugars, alcohols, acids, and certain amino acids. These findings are accompanied by compositional divergence, which reflects the ongoing biochemical transformations. Notwithstanding the absence of overt geographic patterns in molecular composition, national preferences for specific adjuncts or fermentation approaches became evident.

The correlation analysis revealed a positive association between the antioxidant capacity and several compounds, particularly phenolic acids (e.g., ellagic, dihydroxybenzoic and salicylic). Conversely, compounds such as 2-phenylethanol and trigonelline exhibited negative correlations, potentially due to their origin in microbial stress responses or oxidative degradation. The findings emphasize the multifactorial nature of wild ale production and maturation, and underscore the value of metabolomics in establishing a correlation between composition and functional properties.

## Figures and Tables

**Figure 1 ijms-26-06993-f001:**
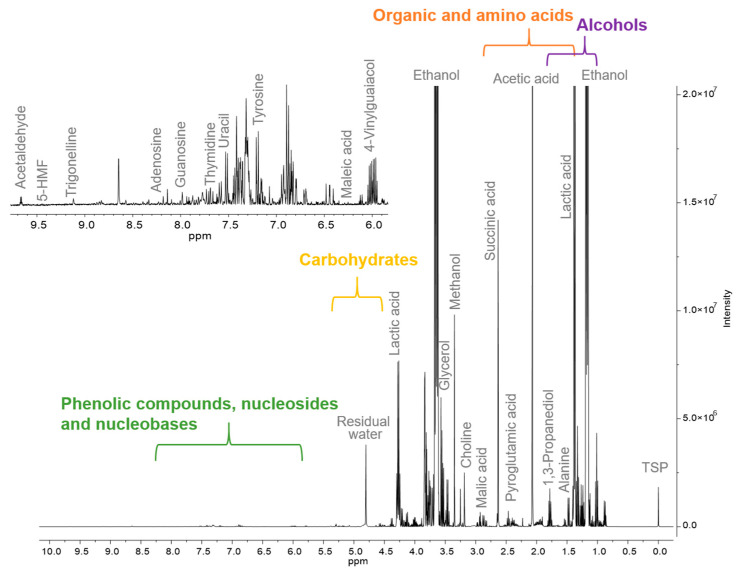
^1^H NMR spectrum of a wild ale (sample W17) with key metabolite assignments.

**Figure 2 ijms-26-06993-f002:**
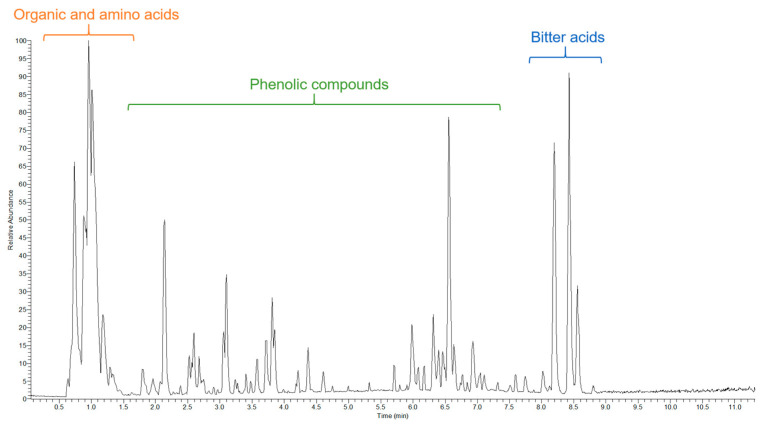
Representative LC-MS chromatogram of a wild ale (sample W17) showing major metabolite groups.

**Figure 3 ijms-26-06993-f003:**
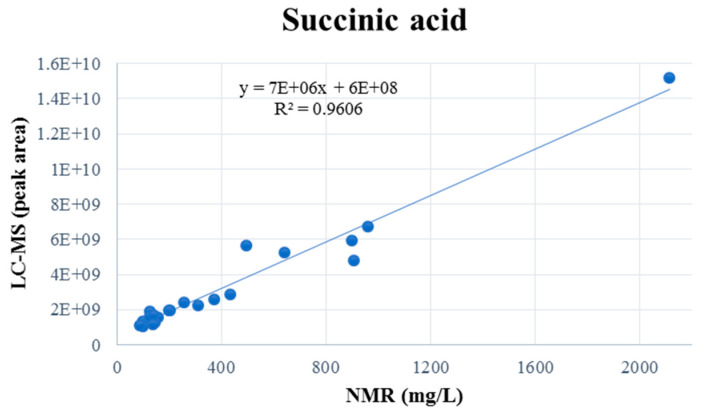
Linear regression comparing succinic acid quantification via NMR and relative intensity data from LC-MS analysis.

**Figure 4 ijms-26-06993-f004:**
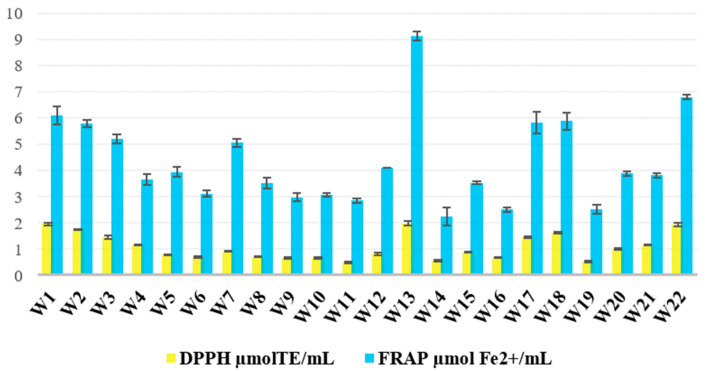
Antioxidant activity of wild ale samples (*n* = 22) assessed via DPPH and FRAP assays.

**Figure 5 ijms-26-06993-f005:**
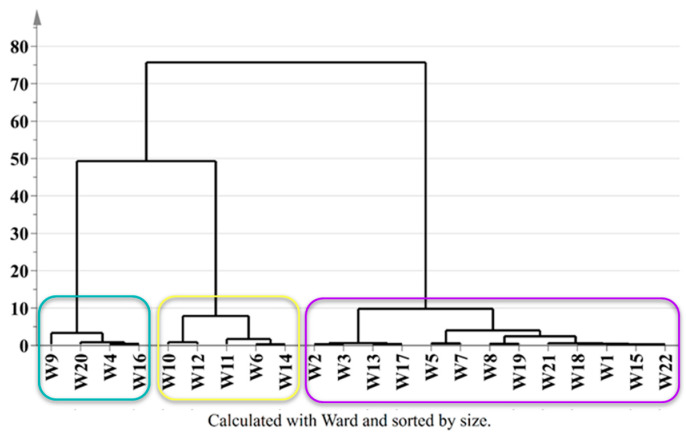
Hierarchical clustering dendrogram of wild ales based on NMR, LC-MS, and antioxidant data, revealing grouping by added ingredients.

**Figure 6 ijms-26-06993-f006:**
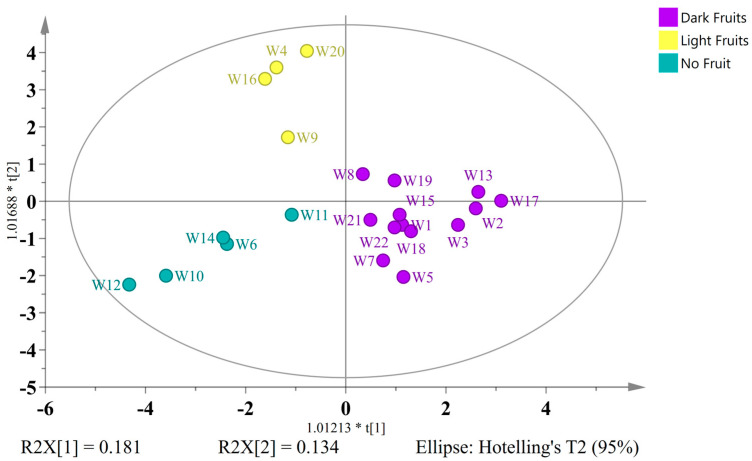
OPLS-DA score plot of wild ales classified by ingredient type—dark fruits (purple), light fruits (yellow), and no fruits (dark teal), using 23 VIP-selected variables (VIP > 0.85).

**Figure 7 ijms-26-06993-f007:**
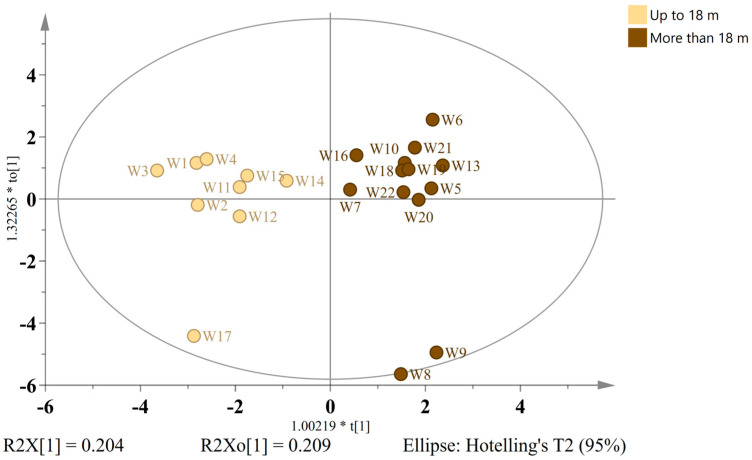
OPLS-DA score plot distinguishing wild ales by aging duration: ≤18 (caramel) vs. >18 months (brown), using 22 discriminatory metabolites.

**Figure 8 ijms-26-06993-f008:**
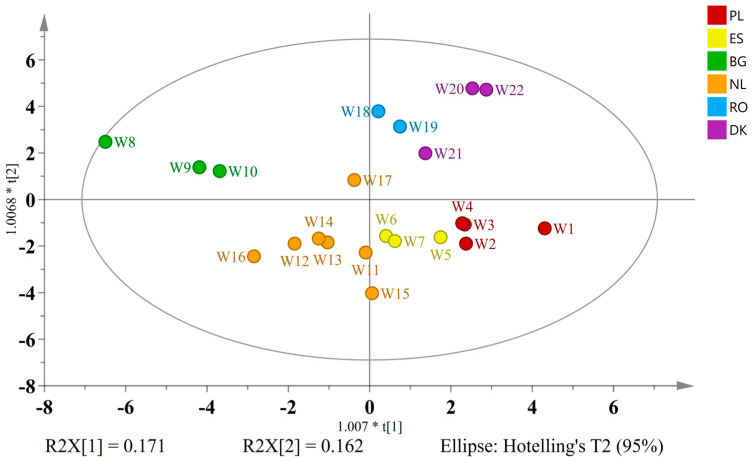
OPLS-DA score plot of wild ales classified by country of origin, using 41 VIP-selected variables (VIP > 0.9).

**Figure 9 ijms-26-06993-f009:**
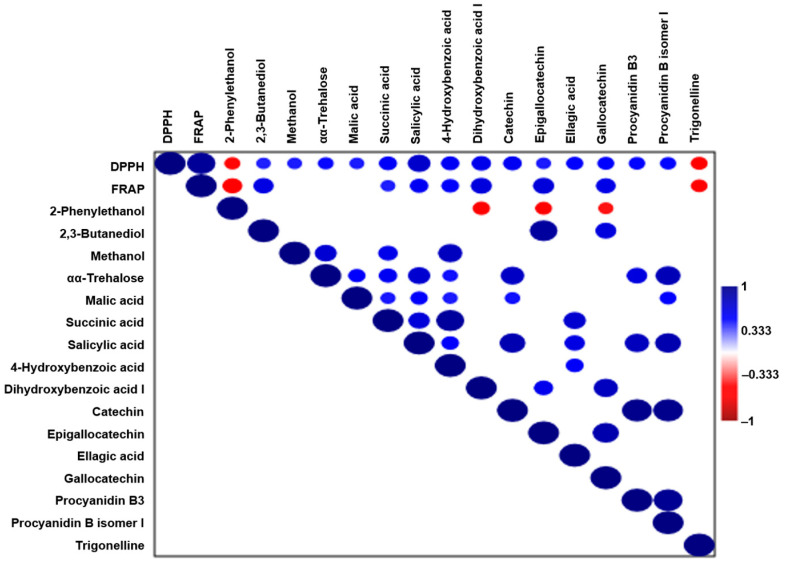
Correlation matrix showing relationships between antioxidant assays (DPPH, FRAP) and key metabolites from NMR and LC-MS analyses.

**Table 1 ijms-26-06993-t001:** List of 53 metabolites, identified and quantified in wild ales via ^1^H NMR spectroscopy—concentrations and chemical shifts.

Compound	Min–Max, *Average*, Median (mg/L)	^1^H δ (ppm), Multiplicities *
** *Alcohols* **		
1-Propanol	13.0–89.0, *32.5*, 23.7	1.54, m
1,3-Propanediol	0.0–1511.4, *183.9*, 7.5	1.78, m
2-Phenylethanol	13.7–57.8, *35.2*, 35.1	7.31, m
2,3-Butanediol	100.3–915.4, *223.5*, 193.1	1.13, d
Ethanol	30651.9–64404.6, *41870.3*, 40965.5	1.17, t
Glycerol	321.6–2848.9, *1650.7*, 1601.7	3.55, dd
Isobutanol	13.5–105.9, *48.2*, 34.7	1.73, m
Isopentanol	42.1–96.9, *60.1*, 55.6	1.43, q
Methanol	2.9–376.2, *59.9*, 21.7	3.35, s
** *Amino acids* **		
Alanine	52.1–426.2, *147.1*, 139.2	1.48, d
Histidine	0.0–2.0, *0.3*, 0.0	7.84, s
Isoleucine	2.9–103.1, *24.9*, 8.5	1.01, d
Leucine	4.3–102.7, *34.9*, 29.3	0.94, d
Phenylalanine	2.8–73.8, *29.8*, 24.8	7.41, m
Pyroglutamic acid	37.5–181.1, *91.4*, 85.7	2.56, m
Tryptophan	3.0–15.8, *8.7*, 8.5	7.26, t
Tyrosine	3.7–54.5, *15.9*, 11.3	7.21, d
Valine	10.5–151.7, *52.4*, 37.5	0.98, d
** *Nucleosides and Nucleobases* **		
Adenosine	0.3–2.6, *0.9*, 0.8	8.37, s
Guanosine	12.8–63.9, *32.3*, 32.6	7.98, s
Inosine	0.5–3.7, *1.4*, 1.2	8.34, s
Thymidine	1.7–16.0, *5.7*, 4.3	7.65, s
Thymine	2.3–10.7, *6.2*, 6.0	1.85, d
Uracil	5.0–36.4, *18.3*, 16.1	7.52, d
Uridine	1.4–53.0, *20.5*, 18.1	7.86, d
** *Organic acids* **		
Acetic acid	413.6–4076.4, *1749.1*, 1494.9	2.07, s
Citric acid	0.0–2075.3, *177.3*, 7.7	2.78, d
Formic acid	0.0–3.7, *0.9*, 0.6	8.31, s
Fumaric acid	0.0–2.3, *0.8*, 0.6	6.59, s
GABA	2.1–36.8, *15.2*, 14.0	3.04, t
Gallic acid	0.5–8.0, *2.8*, 2.4	7.15, s
Lactic acid	3093.6–8986.2, *5915.3*, 5955.8	1.38, d
Maleic acid	0.2–1.3, *0.6*, 0.5	6.33, s
Malic acid	12.3–163.3, *53.5*, 44.1	2.95, dd
Succinic acid	85.4–2115.4, *409.8*, 199.8	2.64, s
Tartaric acid	0.6–886.1, *124.2*, 12.7	4.56, s
** *Saccharides* **		
αα-Trehalose	4.2–79.9, *22.5*, 16.4	5.18, d
Arabinose	19.8–233.0, *71.8*, 60.9	4.50, d
Kojibiose	12.2–797.7, *96.3*, 43.4	5.08, d
Lactose	0.0–3778.3, *177.5*, 0.0	4.43, d
Maltodextrin	474.4–3806.8, *1597.8*, 1354.2	3.26, dd
Mannose	0.0–41.8, *9.7*, 8.2	5.17, d
Raffinose	0.0–59.2, *10.9*, 8.3	5.00, d
Xylose	9.0–64.6, *35.0*, 35.9	5.17, d
** *Other organic compounds* **		
4-Vinylguaiacol	0.0–62.0, *6.1*, 0.0	6.00, ddt
5-Hydroxymethylfurfural	0.1–1.6, *0.6*, 0.6	9.44, s
Acetaldehyde	0.2–7.3, *1.3*, 0.7	9.67, q
Acetoin	4.5–35.2, *17.0*, 14.5	2.23, s
Betaine	39.1–133.0, *83.7*, 82.1	3.25, s
Choline	40.8–150.9, *77.9*, 74.5	3.18, s
Ethyl acetate	17.2–238.0, *88.0*, 35.9	4.13, q
Isobutyraldehyde	0.0–0.1, *0.0*, 0.0	9.61, s
Trigonelline	3.0–11.9, *6.3*, 5.7	9.12, s

* Multiplicities: s—singlet, d—doublet, t—triplet, dd—doublet of doublets, q—quartet, ddt— doublet of doublet of triplets, m—multiplet.

**Table 2 ijms-26-06993-t002:** List of 62 compounds, identified via LC-MS in wild ales, including retention times (RTs), molecular formula, mass error, exact mass (*m*/*z*), MS/MS fragments, and identification level.

No.	Compound	RT, Min	Molecular Formula	Δ ppm	[M−H]^−1^	MS/MS Fragments	Identification
1	L-Aspartic acid	0.71	C_4_H_7_NO_4_	−9.03	132.0290		Standard
2	L-Glutamic acid	0.73	C_5_H_9_NO_4_	−6.14	146.0449		Standard
3	D-(−)-Quinic acid	0.78	C_7_H_12_O_6_	−4.61	191.0552		Standard
4	L-(+)-Tartaric acid	0.78	C_4_H_6_O_6_	−7.81	149.0080	149, 87, 73, 59	NMR
5	L-Malic acid	0.81	C_4_H_6_O_5_	−8.98	133.0131		NMR
6	Citric acid	1.00	C_6_H_8_O_7_	−3.90	191.0189		Standard
7	Succinic acid	1.21	C_4_H_6_O_4_	−11.20	117.0180	117, 73	NMR
8	Gallic acid	1.45	C_7_H_6_O_5_	−5.58	169.0133		Standard
9	Gallocatechin	2.22	C_15_H_14_O_7_	1.35	305.0671	305, 125	[32]
10	Protocatechuic acid	2.29	C_7_H_6_O_4_	−6.60	153.0183		Standard
11	Catechin-*O*-hexoside	2.44	C_21_H_24_O_11_	1.75	451.1254	451, 289	[33]
12	Caftaric acid	2.51	C_13_H_12_O_9_	1.28	311.0413		Standard
13	Neochlorogenic acid	2.55	C_16_H_18_O_9_	0.87	353.0881		Standard
14	Procyanidin B isomer I	2.83	C_30_H_26_O_12_	1.75	577.1362	577, 407, 289, 125	[32]
15	(−)-Epigallocatechin	2.88	C_15_H_14_O_7_	1.26	305.0671		Standard
16	4-Hydroxybenzoic acid	2.97	C_7_H_6_O_3_	−8.22	137.0233		Standard
17	Procyanidin B3	2.97	C_30_H_26_O_12_	1.42	577.1360		Standard
18	Dihydroxybenzoic acid I	2.98	C_7_H_6_O_4_	−6.49	153.0183	153, 109, 108	[34]
19	Chlorogenic acid	3.10	C_16_H_18_O_9_	1.23	353.0882		Standard
20	(+)-Catechin	3.11	C_15_H_14_O_6_	0.97	289.0720		Standard
21	4-Hydroxyphenylacetic acid	3.13	C_8_H_8_O_3_	−5.30	151.0390	151, 107	[33]
22	Procyanidin B isomer II	3.39	C_30_H_26_O_12_	1.86	577.1362	577, 407, 289, 125	[32]
23	Caffeic acid	3.40	C_9_H_8_O_4_	−4.31	179.0341		Standard
24	Dihydroxybenzoic acid II	3.49	C_7_H_6_O_4_	−6.25	153.0184	153, 109	[34]
25	Caffeoylquinic acid	3.59	C_16_H_18_O_9_	1.18	353.0882	191, 85	[33]
26	Ferulic acid-*O*-hexoside I	3.70	C_16_H_20_O_9_	1.24	355.1040	295, 235, 193, 175, 134	[33]
27	Ferulic acid-*O*-hexoside II	3.88	C_16_H_20_O_9_	1.33	355.1039	295, 235, 193, 175, 134	[33]
28	Quercetin 3-sophoroside	3.91	C_27_H_30_O_17_	2.25	625.1424		Standard
29	Kaempferol hexoside	4.02	C_21_H_20_O_11_	1.52	447.0940	447, 357, 327, 285, 133	[32]
30	Myricetin 3-galactoside	4.08	C_21_H_20_O_13_	1.71	479.0839		Standard
31	*trans*-*p*-Coumaric acid	4.12	C_9_H_8_O_3_	−5.14	163.0391		Standard
32	Orientin (Luteolin 8-*C*-glucoside)	4.16	C_21_H_20_O_11_	1.70	447.0940		Standard
33	Rutin	4.43	C_27_H_30_O_16_	1.87	609.1473		Standard
34	Ellagic acid	4.44	C_14_H_6_O_8_	0.69	300.9992		Standard
35	Procyanidin B isomer III	4.51	C_30_H_26_O_12_	2.40	577.1365	407, 289, 125	[32]
36	Hyperoside	4.53	C_21_H_20_O_12_	1.91	463.0891		Standard
37	Quercetin 3-glucoside	4.60	C_21_H_20_O_12_	1.84	463.0891		Standard
38	Kaempferol 3-rutinoside	4.83	C_27_H_30_O_15_	2.12	593.1525		Standard
39	Guajaverin (Quercetin 3-*O*-α-L-arabinopyranoside)	4.85	C_20_H_18_O_11_	1.37	433.0782		Standard
40	3,5-Dicaffeoylquinic acid	4.98	C_25_H_24_O_12_	1.26	515.1202		Standard
41	Kaempferol 3-glucoside	5.01	C_21_H_20_O_11_	1.69	447.0941		Standard
42	4,5-Dicaffeoylquinic acid	5.22	C_25_H_24_O_12_	1.56	515.1203		Standard
43	Hesperidin	5.25	C_28_H_34_O_15_	1.36	609.1833	609, 301, 151	[34]
44	Rosmarinic acid	5.34	C_18_H_16_O_8_	1.43	359.0778		Standard
45	Sallicylic acid (2-hydroxybenzoic acid)	5.36	C_7_H_6_O_3_	−8.44	137.0233		Standard
46	Myricetin	5.36	C_15_H_10_O_8_	0.86	317.0306	317, 179, 151, 137	[35]
47	Luteolin	5.96	C_15_H_10_O_6_	0.87	285.0407		Standard
48	Quercetin	5.98	C_15_H_10_O_7_	0.63	301.0356		Standard
49	Resveratrol	6.09	C_14_H_12_O_3_	−0.93	227.0712	227, 185, 143	[35]
50	Naringenin (4′,5,7-Trihydroxyflavanone)	6.32	C_15_H_12_O_5_	0.89	271.0614		Standard
51	Apigenin	6.33	C_15_H_10_O_5_	1.16	269.0459		Standard
52	Kaempferol	6.38	C_15_H_10_O_6_	0.77	285.0407		Standard
53	Chrysoeriol (Luteolin 3′-methyl ether)	6.42	C_16_H_12_O_6_	0.81	299.0563		Standard
54	Isorhamnetin	6.47	C_16_H_12_O_7_	1.18	315.0514		Standard
55	Isoxanthohumol	6.77	C_21_H_22_O_5_	1.22	353.1399		Standard
56	Desmethylxanthohumol	7.70	C_20_H_20_O_5_	0.86	339.1241	339, 219, 119	[36]
57	Xanthohumol	7.92	C_21_H_22_O_5_	1.49	353.1400		Standard
58	iso-α-Cohumulone	8.19	C_20_H_28_O_5_	0.63	347.1866	347, 329, 251, 181	[33]
59	iso-α-ad/*n*-humulone	8.42	C_21_H_30_O_5_	1.15	361.2025	361, 343, 265, 195	[33]
60	iso-α-ad/*n*-humulone	8.55	C_21_H_30_O_5_	1.23	361.2025	361, 343, 265, 195	[33]
61	Cohumolone	8.78	C_20_H_28_O_5_	1.06	347.1868	347, 278, 235, 207	[33]
62	*n*-Humulone	8.94	C_21_H_30_O_5_	1.56	361.2026	361, 343, 292, 265	[33]

## Data Availability

The original contributions presented in this study are included in the article/Appendix A. The data presented in this study are available on reasonable request from the corresponding author.

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
