# Peer review of "Integrated Spectroscopic Analysis of Wild Beers: Molecular Composition and Antioxidant Properties"

_ijms, 2025, doi:10.3390/ijms26146993_

Round 1
Reviewer 1 Report
Comments and Suggestions for Authors
The authors presented a very interesting study on the chemical profile of 22 wild ales. They thoroughly described the materials and methods, and the extensive introduction encourages the reader to deepen their knowledge of the subject. The authors used a variety of methods for analysis. The description of the results is clear and supported by figures and tables. The discussion is perfectly woven into the work results.
However, the paper lacks a paragraph at the end about the limitations of the work and what distinguishes it from other studies on this topic.
The manuscript is missing lists of abbreviations.
The authors analyzed 22 wild ales from six countries. This is a relatively small sample size. Doesn't this number of samples significantly impact the results?
Author Response
The authors presented a very interesting study on the chemical profile of 22 wild ales. They thoroughly described the materials and methods, and the extensive introduction encourages the reader to deepen their knowledge of the subject. The authors used a variety of methods for analysis. The description of the results is clear and supported by figures and tables. The discussion is perfectly woven into the work results.
We would like to thank the reviewer for his positive evaluation and the thoughtful and constructive comments. We are grateful for the acknowledgment of our methodological approach, the clarity of our presentation, and the way in which we integrated our results with the discussion.
However, the paper lacks a paragraph at the end about the limitations of the work and what distinguishes it from other studies on this topic.
We agree with this valuable recommendation. We have added a paragraph to the end of the “Results and Discussion” section titled „Limitations and Perspectives“, which addresses limitations such as sample size and batch variation. This study is differentiated from previous ones by the integration of NMR, LC-MS, and antioxidant assays across multiple countries, ingredients, and aging.
“2.8 Limitations and Future Perspectives
This study provides the most comprehensive metabolomic characterization of wild ales to date by integrating NMR, LC-MS, and antioxidant assays across 22 samples from six countries. While it offers new insights into ingredient- and region-specific signatures, several limitations, including those detailed in Section 2.6, merit consideration. Although the sample size is diverse and covers key brewing regions, it may not capture the full variability of global production. Additionally, differences in spontaneous fermentation (e.g., microbial community variability) and barrel-aging conditions (e.g., wood type and reuse history) between batches could affect reproducibility. Our approach surpasses previous GC-MS-focused beer metabolomics studies by capturing both volatile and non-volatile compounds and linking them to functional antioxidant properties. Unlike earlier research emphasizing microbial succession, we relate chemical profiles to specific production variables, such as fruit/herb additions, aging duration, and geography, to support authenticity assessment. Identifying biomarkers associated with ingredient use and aging duration provides practical tools for quality control. These advances lay the groundwork for future research involving larger datasets and the integration of adjunct dosing control, microbial-metabolite network analysis, and sensory data on consumer preferences. The potential role of hop-derived compounds in shaping microbial stability and fermentation dynamics is also an important area for future investigation.”
The manuscript is missing lists of abbreviations.
We thank the reviewer for this important observation. We have now added a “List of Abbreviations” section after the “Conclusions”. This includes all abbreviations used in the manuscript, including NMR, LC-MS, DPPH, FRAP, OPLS-DA and VIP.
The authors analyzed 22 wild ales from six countries. This is a relatively small sample size. Doesn't this number of samples significantly impact the results?
We appreciate this important point. We've addressed it in the new “Limitations and Future Perspectives” paragraph, where we note that the sample size may limit generalizability of the findings and recommend the use of larger and more balanced datasets.

Reviewer 2 Report
Comments and Suggestions for Authors
In this manuscript, the authors performed metabolomic profiling based on 1H NMR and LC-MS analyses and antioxidant activity determinations (FRAP and DPPH), combined with detailed chemometric analysis. These methods constitute a reliable set of analytical tools allowing for determining the effects of fruit and herb additions, fermentation duration, and geographic origin on the metabolomic profile and antioxidant activity of wild ales. The methods applied are appropriate, and the conclusions are supported by the results. Additionally, this research fills the gap regarding wild ales. Overall, I recommend this work for publication in International Journal of Molecular Sciences after minor revisions:
Line 237: you should list what compounds you mean
Subsection 2.3. Antioxidant Capacity of Wild Ales
The results of DPPH assay were expressed as µmol Trolox equivalents per mL, whereas the results of FRAP assay were expressed as µmol Fe²⁺ equivalents per mL. To effectively compare both methods, the results obtained using the FRAP method should also be expressed in µmol Trolox equivalents per mL (Figure 4).
Table 2, number 43 (hesperidin), line 271, and lines 466-467, the following appears: Error! Bookmark not defined. The errors should be corrected.
Author Response
In this manuscript, the authors performed metabolomic profiling based on 1H NMR and LC-MS analyses and antioxidant activity determinations (FRAP and DPPH), combined with detailed chemometric analysis. These methods constitute a reliable set of analytical tools allowing for determining the effects of fruit and herb additions, fermentation duration, and geographic origin on the metabolomic profile and antioxidant activity of wild ales. The methods applied are appropriate, and the conclusions are supported by the results. Additionally, this research fills the gap regarding wild ales. Overall, I recommend this work for publication in International Journal of Molecular Sciences after minor revisions:
We would like to thank the reviewer for their positive assessment of our manuscript and for highlighting its methodological rigor and contribution to our understanding of wild ales. We are also grateful for the careful reading and constructive suggestions.
Line 237: you should list what compounds you mean
We thank the reviewer for this helpful observation. In line 237, we referred to the compounds that were detected using both NMR and LC-MS. To improve clarity, we have added the following explicit list to the text “five compounds —succinic, citric, malic, tartaric, and gallic acids—were detected”.
Subsection 2.3. Antioxidant Capacity of Wild Ales
The results of DPPH assay were expressed as µmol Trolox equivalents per mL, whereas the results of FRAP assay were expressed as µmol Fe²⁺ equivalents per mL. To effectively compare both methods, the results obtained using the FRAP method should also be expressed in µmol Trolox equivalents per mL (Figure 4).
We appreciate this thoughtful comment. We agree that expressing both assays in the same unit (e.g., µmol Trolox equivalents) would enable direct comparison. However, due to the iron-reducing reaction mechanism, FRAP assays are typically calibrated against Fe²⁺ standards, which differ chemically from the radical-scavenging reaction of DPPH, which is calibrated with Trolox. As the FRAP method is standardized to Fe²⁺ equivalents in the literature and as our calibration followed this conventional approach, we suggest maintaining the current units to ensure methodological consistency with previous studies.
Table 2, number 43 (hesperidin), line 271, and lines 466-467, the following appears: Error! Bookmark not defined. The errors should be corrected.
We thank the reviewer for pointing out these formatting errors. We have carefully reviewed the manuscript and corrected all instances of “Error! Bookmark not defined” in Table 2 and the main text (lines 271, 466–467, and elsewhere). All references are now correctly formatted.

Reviewer 3 Report
Comments and Suggestions for Authors
Basic chemical parameters should be included in Table S1.
Line 63 The wine?
Line 162 Median and outliers should be shown in Table 1.
Line 221 The concentration of 62 compounds should be provided.
Line 240 The same figure of the other four compounds should also be shown here.
Line 308-311 Significance analysis is necessary.
Figure 6, 7 and 8 The selected variables should also be shown, then it will be easier to know the valuable indicators.
Author Response
Basic chemical parameters should be included in Table S1.
We thank the reviewer. As these beers were produced by independent craft breweries, the only basic chemical parameter that was consistently disclosed was ABV%. This information is now included in Table S1 of the Supplementary Materials.
Line 63 The wine?
We thank the reviewer for pointing out this typographical error. The original sentence read: “Maturation serves to enhance the chemical and sensory intricacy of the wine”. In the revised manuscript, we have replaced “the wine” with “the beer”.
Line 162 Median and outliers should be shown in Table 1.
We appreciate this helpful suggestion. To improve the statistical presentation, we have added median values to Table 1, alongside the existing minimum, maximum, and average. Including the median provides a robust measure of central tendency that is less affected by outliers. However, we have chosen not to include IQR, as this would increase the complexity of Table 1 without adding proportionate interpretive value. We believe that this addresses the reviewer's concern about illustrating data spread while maintaining readability.
Line 221 The concentration of 62 compounds should be provided.
We thank the reviewer for this important comment. We would like to clarify that the design of our LC-MS analysis was either untargeted or semi-targeted. As such, we did not determine absolute concentrations, rather, we used relative intensities. We have added a sentence to the “Materials and Methods” section that explicitly states this limitation: “This untargeted LC-MS approach provided qualitative or semi-quantitative identifications without determining absolute concentrations.” Therefore, Table 2 only reports identification details.
Line 240 The same figure of the other four compounds should also be shown here.
We appreciate this suggestion. However, to avoid overloading the main text, we have decided to keep these regression plots in the Supplementary Materials, where they are clearly referenced and available.
Line 308-311 Significance analysis is necessary.
We agree that including a formal significance analysis would strengthen this section. We have now conducted one-way ANOVA tests to compare antioxidant capacity (DPPH and FRAP values) between ingredient-based groups (dark fruits, light fruits and no fruits) and aging duration groups (≤18 months and >18 months). Significant differences were observed for both the DPPH (F = 4.996, p = 0.018) and the FRAP (F = 5.199, p = 0.015) tests between the ingredient groups. However, no significant differences were found between the aging duration groups (DPPH: F = 0.439, p = 0.514; FRAP: F = 0.003, p = 0.951). These results have been explicitly added to Section 2.3 to support the interpretation.
Figure 6, 7 and 8 The selected variables should also be shown, then it will be easier to know the valuable indicators.
We thank the reviewer for this valuable suggestion. To improve the interpretability of the OPLS-DA models in Figures 6, 7, and 8, we have added loading and VIP plots , which are provided as Supplementary Figures S3 and S4.

Reviewer 4 Report
Comments and Suggestions for Authors
Integrated Spectroscopic Analysis of Wild Beers: Molecular Composition and Antioxidant Properties
Dessislava Gerginova et al.
Among the beer categories that are markedly appealing to all involved in the sector are so-called wild ales or spontaneously fermented beers. The fermentation processes that are rather distinct from one beer to another are mainly influenced by microbial populations leading to a vast array of complex properties relating to taste, aroma and flavor.
The current work was based on an integrated metabolomic profiling procedure combining proton nuclear magnetic resonance (¹H NMR) spectroscopy, liquid chromatography–mass spectrometry (LC-MS), and spectrophotometric assays (DPPH and FRAP) to characterize the molecular composition and antioxidant potential of 22 wild ales from six countries. Among the total of 53 compounds quantified by NMR and 62 by LC-MS, organic acids, amino acids, sugars, alcohols, bitter acids, phenolic compounds were prominently present. Prolonged ageing (>18 months) was associated with increased levels of certain amino acids, fermentation-derived aldehydes, and phenolic degradation products. Moreover, addition of dark fruits with herb-infusions showed divergent phytochemical and functional profiles. The appearance of specific metabolites indicated the influence of regional brewing practices on beer composition. Furthermore, the major contributors to antioxidant activities were identified as salicylic, dihydroxybenzoic, and 4-hydroxybenzoic acids.
***
The work presented here proves that the great scientific endeavors delivered by the Bulgarian research team have revealed complex chemical signatures across the wild ale samples by targeted application of nuclear magnetic resonance and liquid chromatography-mass spectrometry metabolomics. Aside the inherent microbial attributes, still the essential role of hops clearly emerged, although the details regarding the influences on microbial stability remain enigmatic for the time being.
Author Response
Integrated Spectroscopic Analysis of Wild Beers: Molecular Composition and Antioxidant Properties
Dessislava Gerginova et al.
Among the beer categories that are markedly appealing to all involved in the sector are so-called wild ales or spontaneously fermented beers. The fermentation processes that are rather distinct from one beer to another are mainly influenced by microbial populations leading to a vast array of complex properties relating to taste, aroma and flavor.
The current work was based on an integrated metabolomic profiling procedure combining proton nuclear magnetic resonance (¹H NMR) spectroscopy, liquid chromatography–mass spectrometry (LC-MS), and spectrophotometric assays (DPPH and FRAP) to characterize the molecular composition and antioxidant potential of 22 wild ales from six countries. Among the total of 53 compounds quantified by NMR and 62 by LC-MS, organic acids, amino acids, sugars, alcohols, bitter acids, phenolic compounds were prominently present. Prolonged ageing (>18 months) was associated with increased levels of certain amino acids, fermentation-derived aldehydes, and phenolic degradation products. Moreover, addition of dark fruits with herb-infusions showed divergent phytochemical and functional profiles. The appearance of specific metabolites indicated the influence of regional brewing practices on beer composition. Furthermore, the major contributors to antioxidant activities were identified as salicylic, dihydroxybenzoic, and 4-hydroxybenzoic acids.
***
The work presented here proves that the great scientific endeavors delivered by the Bulgarian research team have revealed complex chemical signatures across the wild ale samples by targeted application of nuclear magnetic resonance and liquid chromatography-mass spectrometry metabolomics. Aside the inherent microbial attributes, still the essential role of hops clearly emerged, although the details regarding the influences on microbial stability remain enigmatic for the time being.
We would like to thank the reviewer for this thoughtful and positive evaluation of our work. We appreciate the comment on the role of hops, and we agree that the extent to which it influences microbial stability and fermentation dynamics remains unclear. While this topic was beyond the scope of our current study, we have noted it as a promising direction for future research in the Results and Discussion section – 2.8.
